# Thermodynamic forces from protein and water govern condensate formation of an intrinsically disordered protein domain

Saumyak Mukherjee [1] & Lars V. Schäfer [1] ✉

Liquid-liquid phase separation (LLPS) can drive a multitude of cellular processes by compartmentalizing biological cells via the formation of dense liquid biomolecular condensates, which can function as membraneless organelles. Despite its importance, the molecular-level understanding of the underlying thermodynamics of this process remains incomplete. In this study, we use atomistic molecular dynamics simulations of the low complexity domain (LCD) of human fused in sarcoma (FUS) protein to investigate the contributions of water and protein molecules to the free energy changes that govern LLPS. Both protein and water components are found to have comparably sizeable thermodynamic contributions to the formation of FUS condensates. Moreover, we quantify the counteracting effects of water molecules that are released into the bulk upon condensate formation and the waters retained within the protein droplets. Among the various factors considered, solvation entropy and protein interaction enthalpy are identified as the most important contributions, while solvation enthalpy and protein entropy changes are smaller. These results provide detailed molecular insights on the intricate thermodynamic interplay between protein- and solvation-related forces underlying the formation of biomolecular condensates.

The interior of a biological cell is densely packed with biomolecules[1,2], with fractions ranging up to 30% in volume or concentrations of 300 mg mL$^{-1}$. In this crowded environment, a multitude of biomolecules interact with each other[1,3–5]. Biological functions can be modulated or even governed by such interactions[6–8]. For example, protein–protein interactions are key for maintaining cellular homeostasis[9], can modulate protein stability[2], and may sometimes also lead to aggregation, with associated complications and diseases such as cataracts, Alzheimer's, Parkinson's, frontotemporal dementia and amyotrophic lateral sclerosis[10–16]. Hence, understanding the nature of these interactions and unraveling the molecular driving forces that are at play in such crowded biomolecular environments is of fundamental importance, as it might form the basis for targeted manipulation[17].

Apart from the biomolecules themselves, their interactions with and coupling to water, another major component of the cell, are also crucial[18–22]. Water is essential for many biomolecular processes, including aggregation/association, and the thermodynamic contributions linked to water can be substantial[12,21,23,24]. A prime example of biomolecular association is liquid–liquid phase separation (LLPS) of proteins and nucleic acids[25–28]. LLPS leads to compartmentalization within biological cells via formation of dense biomolecular condensates or droplets[29,30], which are dispersed in a more dilute environment. This process depends on a multitude of factors such as temperature, pressure, pH, cosolvents, salt concentration, etc.[28,31].

Membraneless organelles are intensely researched because they have been shown to serve as selective microreactors within cells in which specific biochemical reactions can take place. These can be pivotal for a plethora of cellular processes, such as RNA splicing, receptor-mediated signaling, and mitosis[32–36]. The dynamic nature and liquid-like character of these condensates allow for the efficient exchange of components with the surroundings[37].

[1]Center for Theoretical Chemistry, Ruhr University Bochum, D-44780 Bochum, Germany. ✉e-mail: lars.schaefer@ruhr-uni-bochum.de

Several proteins have been shown to undergo LLPS[38,39]. The fused in sarcoma (FUS) RNA-binding protein is one such protein that was found to form intracellular condensates[40–42]. FUS is crucial for RNA shearing and transport, DNA repair, micro-RNA processing, gene transcription, and regulation[43–47]. Human FUS is a 525-residue protein with an N-terminal intrinsically disordered low complexity domain (LCD). This LCD region is responsible for the LLPS of FUS[48–50], primarily mediated by multivalent interactions.

Solvation effects are important for protein condensate formation[23]. The present work is based on the consideration that when proteins come close to each other in the condensate, some of the water molecules in the vicinity of the protein surfaces are replaced by other protein moieties. Hence, these water molecules are released into the surrounding dilute phase, resulting in a partial dewetting of the protein surfaces. It was speculated that this water release could be associated with an entropy gain[23]. Indeed, experimental phase transition data for the N-terminal part of Ddx4[51] and for tau-RNA droplet formation[52], analyzed within the framework of the Flory-Huggins model of polymer phase transitions, suggests that the phase separation entropy was favorable. Recent THz spectroscopy experiments were interpreted along the same lines[53,54]. However, at the same time, the condensates also retain a substantial amount of water[49,55], which could experience entropy loss due to increased confinement. Figure 1 schematically illustrates the idea of the interplay, or tug-of-war, of released and retained water, which also forms the basis of the present work.

Due to the challenges associated with studying such fluctuating biomolecular condensates at the required resolution in space and time, an atomic-level picture that connects the structure and dynamics of protein condensates with molecular thermodynamics is largely lacking. Atomistic molecular dynamics (MD) simulations with explicit solvent can, in principle, provide such detailed insights into the structure, composition, and dynamics of protein condensates. However, the huge computational cost renders it impossible to access the slow timescales needed to observe the process of protein condensate formation. In addition, very large simulation systems are required to accommodate the two coexisting phases, which further increases the computational effort. To the best of our knowledge, there have been only few atomistic MD simulation studies of biomolecular condensates[56–60], none of which simulated the actual process of phase separation. Phase separation processes, however, have been simulated using coarse-grained descriptions that are computationally efficient and greatly accelerate the simulations[49,61–65]. However, such coarse-grained models have limitations concerning their ability to provide accurate, atomically-detailed thermodynamic and dynamic evaluations of the protein and, in particular, of the solvent, which is the scope of the present work.

Experimentally, protein condensates are challenging to study with high resolution. One particular challenge also includes the accurate measurement of the protein concentration in the condensate. McCall et al.[66] have used quantitative phase microscopy to determine the condensate concentration of a full-length FUS protein (tagged with a fluorescent protein) to be $337 \pm 8 \, \text{mg mL}^{-1}$. Murakami et al. have recently used Raman imaging microscopy to determine the concentration of proteins in droplets formed as a result of LLPS[67]. This method has also been applied on FUS-LCD, yielding condensate and dilute phase concentrations of a recombinant FUS-LCD of 15 mM (320 mg mL$^{-1}$) and ~60 μM (1 mg mL$^{-1}$), respectively, under the experimental conditions (room temperature, physiological salt concentration)[48]. Another study by Murthy et al.[55] reported the condensate concentration to be 477 mg mL$^{-1}$ for FUS-LCD. The concentration range obtained from the above experiments is quite broad, but they nevertheless provide a concentration window that would represent a FUS condensate.

As simulating the actual process of phase separation is currently impossible with atomistic MD simulations, knowing the concentrations of the dilute and dense phases of FUS-LCD is required for extracting the thermodynamic contributions underlying condensate formation from MD simulations. Since free energy is a state function, separate simulations of (a set of) homogeneous FUS-LCD solutions, including the final protein condensate concentration (Fig. 2), and of the dilute phase yield access to the states necessary for computing the thermodynamic changes associated with the process. For the FUS-LCD condensate concentration, we used a concentration of 350 mg mL$^{-1}$. For the dilute phase, we consider bulk water because the concentration of FUS-LCD is very low (1 mg mL$^{-1}$, see above)[48].

Here, we unravel the thermodynamic driving forces underlying the formation of FUS-LCD condensates using atomistic MD simulations, which enable the decomposition of the total free energy change into contributions from protein–protein, protein–water, and water–water interactions. The analyses provide insights into the entropy and enthalpy changes associated with each of these components. A tug-of-war between the entropy of the released and the retained water molecules is found to be a major mechanistic determinant in the process. In addition, favorable protein–protein interactions also contribute substantially to the thermodynamic driving forces. Entropy-dominated solvation effects and enthalpy-dominated protein interactions are found to be comparable in magnitude, supporting the notion that for a complete and quantitative picture, both effects need to be considered.

## Results and discussion
### Solvent rearrangements upon condensate formation
Protein condensates are formed via LLPS when the concentration reaches a threshold value (which can vary depending on conditions such as temperature, pressure, and other external factors). Upon condensation, the protein molecules are brought into close proximity to each other until eventually, at high concentrations, the

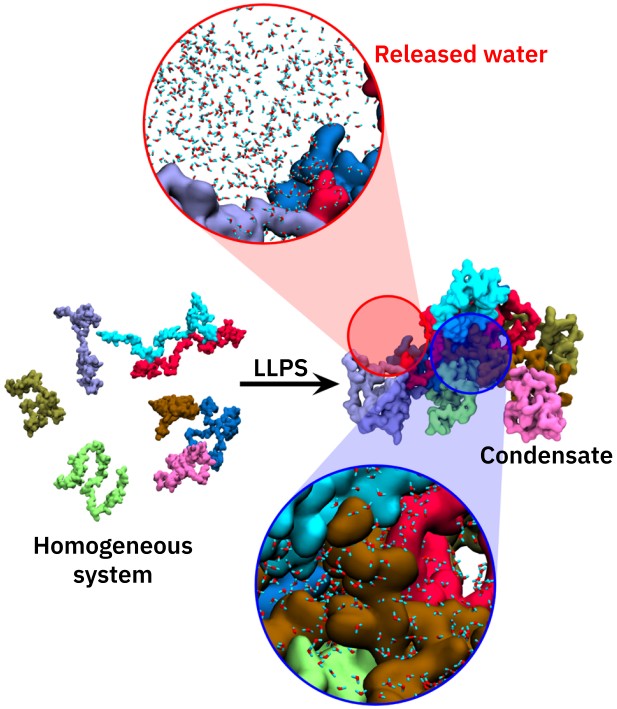

**Fig. 1 | Illustration of the process of protein condensate formation via liquid–liquid phase separation.** The zoomed-in views highlight the water molecules that are released into a bulk-like environment (top) outside the protein condensate and the ones that are retained inside the condensate (bottom).

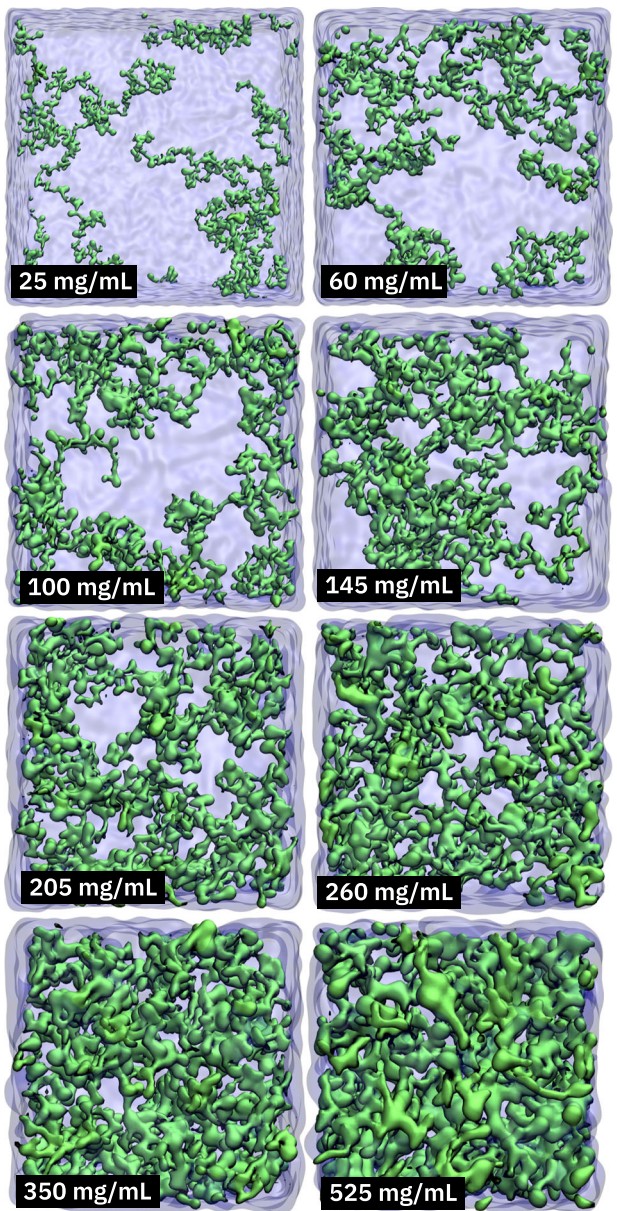

**Fig. 2 | Snapshots of simulated systems.** Each of the simulated systems contains eight FUS-LCD molecules (residues 1–163 of human FUS).

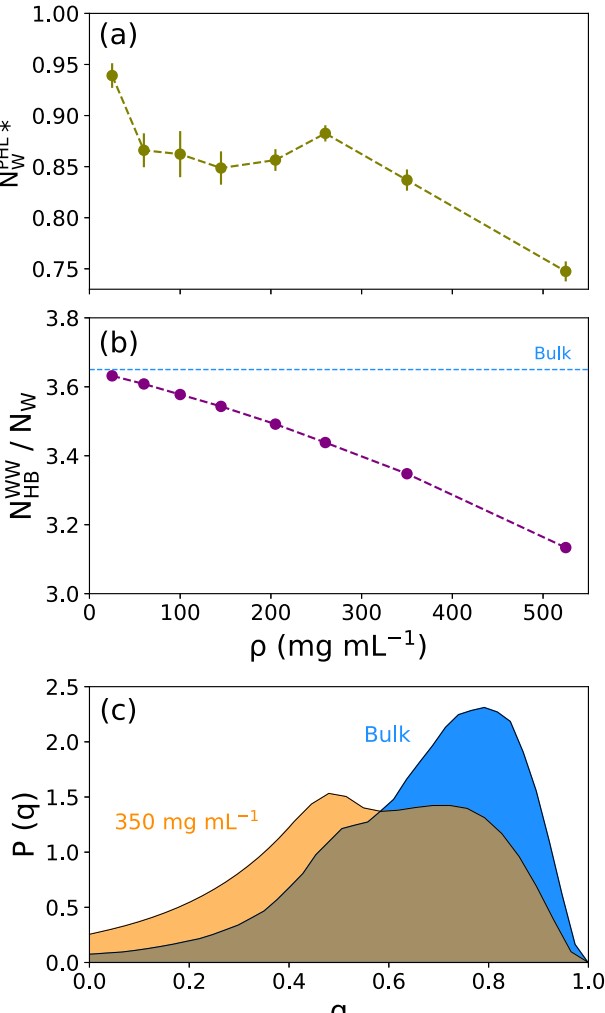

**Fig. 3 | Hydration properties at different FUS-LCD concentrations. a** The number of water molecules in the protein hydration layer (defined here as water molecules within 0.3 nm of the protein surface). $N_W^{PHL}$ is plotted against the protein concentration $\rho$. The asterisk (*) denotes that the numbers are normalized with respect to the number of hydration waters found for a single FUS-LCD protein in the high-dilution limit. **b** The number of water-water hydrogen bonds per water molecule is plotted as a function of $\rho$. **c** Tetrahedral order parameter distribution of water in 350 mg mL$^{-1}$ FUS-LCD solution (orange) compared to bulk water (blue). Data are presented as mean values ± standard deviation (SD) over three repeat simulations (the statistical errors in (**b**) are smaller than the size of the dots). Source data are provided as a Source Data file.

protein hydration layers (PHLs) start to overlap and are eventually even partially stripped off. Consequently, the properties of the water that is confined inside a dense condensate are expected to be affected and to be distinctly different compared to the dilute solution. As shown in Fig. 1, some of the waters are released from the condensate into the dilute phase, while others remain inside the droplets.

To characterize the changing hydration water populations upon condensate formation, the number of water molecules within 0.3 nm from the protein surface ($N_W^{PHL}$) was analyzed, that is, we focused on the water molecules in the first hydration shell that are in direct contact with the protein. In Fig. 3a, the number of hydration waters per protein molecule is plotted as a function of the protein concentration. The numbers in this plot are normalized with respect to the number of water molecules in the hydration layer of a simulation system with only a single FUS-LCD molecule, that is, under dilute conditions. Even at 25 mg mL$^{-1}$, the PHL has 6% fewer water molecules than at infinite dilution. As expected, $N_W^{PHL}$ decreases

with increasing protein concentration. The decrease in hydration water population is due to the release of some water molecules from the PHL into the dilute phase in the course of condensation. However, the proteins in the condensate retain a substantial fraction of their first shell PHL water molecules even at high concentrations.

The spatial confinement imposed by the high protein concentration affects the hydrogen bond (HB) network between the water molecules. With increasing protein concentration, perturbations in the HB network result in the loss of water–water HBs. Besides the fact that the water concentration itself is decreased in the condensates, the waters that are retained inside the condensate form HBs with the protein. This results in a decrease in the number of water–water HBs formed per water molecule (Fig. 3b). This spatial confinement effect is also reflected in the tetrahedral arrangement of water molecules, with decreasing population of water molecules in a tetrahedral local environment with increasing $\rho$. The local environment around a water

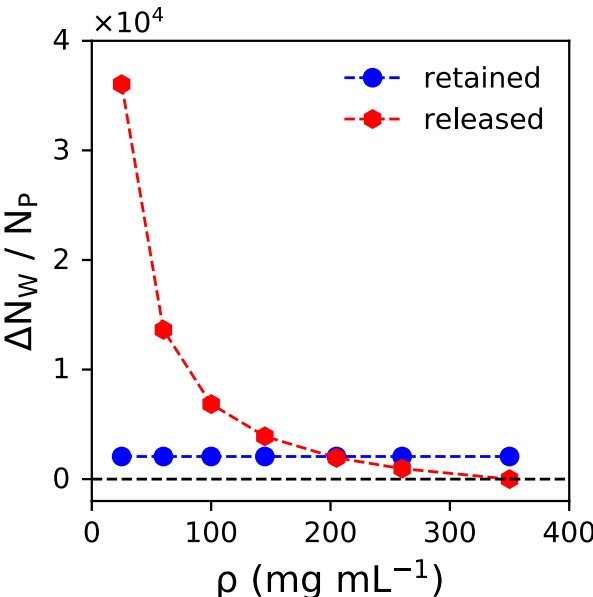

**Fig. 4 | Number of released and retained water molecules per protein.** The number of released molecules decreases (red dashed line) as the FUS-LCD concentration $\rho$ approaches the condensate concentration of 350 mg mL$^{-1}$, where every protein is solvated (on average) by 2064 water molecules (blue dashed line), which are referred to as the retained waters. Source data are provided as a Source Data file.

molecule can be quantified by the tetrahedral order parameter (Eq. (1))[68–70]

$$q = 1 - \frac{3}{8} \sum_{j=1}^{3} \sum_{k=j+1}^{4} \left( \cos\psi_{jk} + \frac{1}{3} \right)^2 \tag{1}$$

where $\psi_{jk}$ denotes the angle between the $j^{th}$, the central, and the $k^{th}$ water molecule. For a perfect tetrahedral arrangement, $q = 1$, whereas $q = 0$ for a completely non-tetrahedral configuration. A comparison between the distributions of $q$ for bulk water and for water in the FUS-LCD condensate (350 mg mL$^{-1}$) shows that the peak around $q = 0.8$ decreases in the concentrated protein solution as compared to bulk water, concomitant with an increase in the peak intensity around $q = 0.5$ (Fig. 3c). Thus, the water molecules retained in the protein condensates experience a less tetrahedral local environment and form more two-dimensional trigonal arrangements to accommodate intermolecular hydrogen bonding.

The change in the population of water surrounding the proteins, as analyzed above, classifies the water molecules in the system into two distinct categories, i.e., released and retained. While the former are transferred into the dilute (bulk-like) phase, the latter have a more confined surrounding inside the condensate. To understand the thermodynamic consequences of these changes in water populations, we next focus on the tug-of-war between the released and the retained water molecules in terms of their enthalpic and entropic contributions to the free energy. To that end, we first quantify the number of the released and retained water molecules.

Starting from a (hypothetical) homogeneous solution with a protein concentration $\rho$, the number of released water molecules is given by the difference in the number of waters in the initial system and in the condensate (Eq. (2))

$$\Delta N_W^{rele}(\rho) = N_W(\rho) - N_W(\rho_{cond}) \tag{2}$$

Equation (2) gives the number of water molecules that would be released into the dilute phase if the protein concentration in the initial

homogeneous system (that is, prior to phase separation) was $\rho$ (see Fig. 1). Since $\rho = \rho_{cond}$ in the condensate, $\Delta N_W^{rele}(\rho_{cond}) = 0$.

In contrast to the released waters, the number of water molecules that are retained in the condensate is constant (Eq. (3)),

$$\Delta N_W^{reta}(\rho) = N_W(\rho_{cond}) \tag{3}$$

assuming that the condensate has a constant (fixed) protein concentration. This is shown graphically in Fig. 4, and the corresponding numbers are given in Supplementary Table 1.

**Thermodynamic signatures of FUS-LCD condensate formation**
Knowing the changes in solvent populations during the FUS-LCD condensation forms the basis for understanding the underlying thermodynamic driving forces. The total free energy change ($\Delta G$) upon condensation can be defined to have two contributions (Eq. (4))

$$\Delta G = \Delta G_P + \Delta G_{solv} \tag{4}$$

One part of the free energy change is directly governed by the proteins themselves ($\Delta G_P$), which includes changes in the protein enthalpies ($\Delta H_P$) and entropies ($\Delta S_P$)

$$\Delta G_P = \Delta H_P - T\Delta S_P \tag{5}$$

$\Delta H_P$ comprises of the intra- ($\Delta E_{intra}$) and intermolecular ($\Delta E_{inter}$) potential energies of the proteins present in the system (Eq. (6)), and hence, refers to the total protein–protein interactions ($\Delta E_{PP}$). Since the systems are hardly compressible, the $p\Delta V$ term can be neglected and the enthalpy difference be accurately approximated by the interaction energy difference,

$$\Delta H_P = \Delta E_{intra} + \Delta E_{inter} = \Delta E_{PP} \tag{6}$$

These energy terms are readily available (within statistical error limits) from the force field interaction energies, averaged over the MD trajectories.

The protein entropy in Eq. (5) is very challenging, particularly for large and flexible molecules like IDPs. The entropic penalty paid by the proteins upon the formation of condensates has two parts. First, the proteins experience an entropy penalty that is associated with the restriction of the overall translational and rotational motions of the proteins in the condensates. These two contributions are hard to separate from each other for IDPs, because their high degree of flexibility renders it challenging to unambiguously decouple overall motions from internal dynamics. However, the free energy change due to the entropy of (de)mixing can be estimated from the concentration ratio, $\Delta G = RT \ln(\rho_{cond}/\rho_{dil})$. With the condensate and dilute phase values of 350 mg mL$^{-1}$ and 1 mg mL$^{-1}$, respectively, one obtains -14.5 kJ mol$^{-1}$ for the associated free energy penalty at 300 K. This contribution was neglected because it is much smaller than the other thermodynamic contributions to the condensate formation (and also much smaller than the statistical uncertainties due to limited sampling in the MD simulations), see below.

By construction, the setup of the simulation systems as homogeneous FUS-LCD solutions (with different concentrations) neglects the energy associated with the formation of an interface upon phase separation. Biomolecular condensates typically have ultralow surface tensions[71] ranging from $\gamma = 0.0001$ mN/m up to ca. 0.5 mN/m (for comparison, the air/water surface tension is 72 mN/m at 298 K). To roughly estimate the energetic penalty of interface formation, we calculated the interface free energy of spherical FUS-LCD droplets (c = 350 mg mL$^{-1}$) with radii from 20 up to 5000 nm, thus covering a broad size range. The resulting free energy penalties per protein molecule are small, with the maximum value of 4 kJ mol$^{-1}$ for a

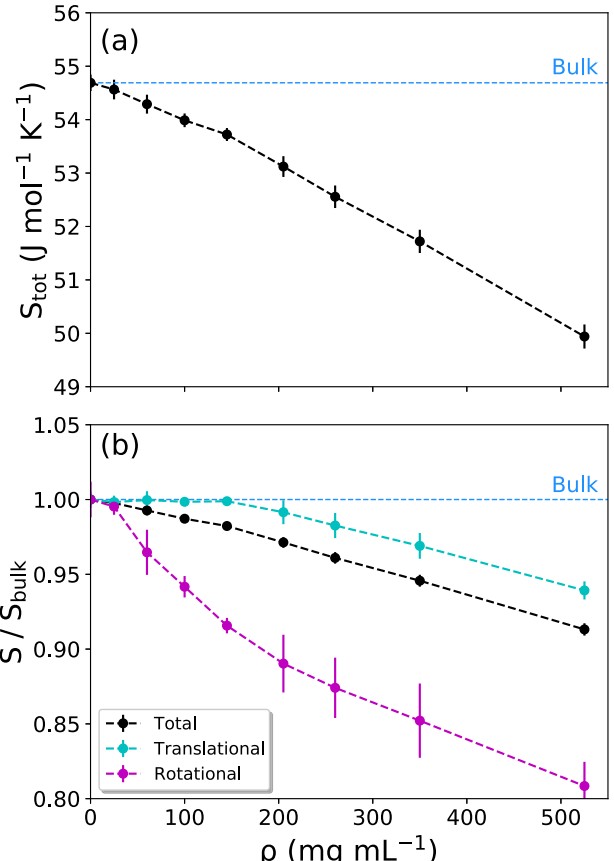

**Fig. 5 | Entropy of water as a function of protein concentration (ρ). a** Total entropy. The horizontal dashed line represents the bulk water value.
**b** Decomposition of the total entropy (black curve) into translational (cyan) and rotational (magenta) contributions. The values in (**b**) were normalized with respect to bulk water. Data in (**a**, **b**) are presented as mean values ± standard deviation (SD) over three repeat simulations. Source data are provided as a Source Data file.

(hypothetical) very small nanodroplet with 20 nm radius and $\gamma = 0.5$ mN/m (which is a factor of more than 150 larger than the actual surface tension of 0.003 mN/m reported for FUS[72]). Therefore, this contribution is neglected in the present work.

Furthermore, one needs to consider the conformational entropy change ($\Delta S_{conf}$) that arises from the differential restriction of the conformational flexibility (and the corresponding changes in the configuration space densities) of the proteins in the condense phase compared to the dilute solution. As is discussed in more detail below, the protein conformational entropy change is notoriously hard to quantify by MD due to sampling limitations, but our estimate indicates that the $-T\Delta S_{conf}$ contribution to the free energy change upon FUS-LCD condensate formation is much smaller than the other, more significant contributions related to solvation and protein–protein interactions.

The second part of the total free energy (Eq. (4)) comes from solvation

$$\Delta G_{solv} = \Delta H_{solv} - T\Delta S_{solv} \tag{7}$$

Note that the index "solv" usually indicates the solvation process defined as the transfer of a solute from the vacuum to the solution, and hence in the present case, where the difference in solvation between protein solutions of different concentrations is considered, the notation $\Delta\Delta G_{solv}$ (or accordingly also for the enthalpy and entropy) might

**Table 1 | Entropy of bulk water at 300 K (a99sb-disp water model) evaluated using the 2PT method**

| Entropy term | Value (J mol⁻¹ K⁻¹) |
|---|---|
| $S_{tot}$ | 54.7 ± 0.2 |
| $S_{tr}$ | 43.8 ± 0.2 |
| $S_{rot}$ | 10.9 ± 0.1 |

The errors denote standard deviation.

be considered more appropriate. However, for simplicity and consistency, we here denote the changes with a single Δ.

Both the solvation enthalpy and entropy in Eq. (7) can be formally divided into protein–water and water–water contributions

$$\Delta H_{solv} = \Delta E_{PW} + \Delta E_{WW} \tag{8}$$

$$\Delta S_{solv} = \Delta S_{PW} + \Delta S_{WW} \tag{9}$$

The enthalpy and entropy terms involving water–water interactions exactly cancel each other (Eq. (10))

$$\Delta E_{WW} - T\Delta S_{WW} = 0 \tag{10}$$

as originally proposed by Ben-Naim[73] and later shown by Yu and Karplus[74] and by Ben-Amotz[75]. Recently, Heinz and Grubmüller clarified the importance of the precise definition of solvent–solvent interactions in this context[76]. From Eqs. (7)–(10), it follows that the total solvation-free energy depends only on the protein–water terms

$$\Delta G_{solv} = \Delta E_{PW} - T\Delta S_{PW} \tag{11}$$

Like in the case of protein–protein interaction energies, $\Delta E_{PW}$ can be directly calculated from the force field energy terms, averaged over the simulation trajectories. However, the entropy of water is more difficult to evaluate due to the diffusive nature of the liquid. We used the 2PT method introduced by Lin et al.[77] for that purpose, see "Methods".

Figure 5a shows that the total molar entropy ($S_{tot}$) of water is decreasing with increasing protein concentration. At higher protein concentrations, the water molecules are located in an increasingly crowded and confined environment that strongly attenuates their configuration space, resulting in lower entropy. A crossover is observed at around $\rho = 150$ mg mL⁻¹ from one linear gradient to another, steeper one (see also Supplementary Fig. 1). The underlying picture becomes clearer when the total entropy is decomposed into its translational ($S_{tr}$) and rotational ($S_{rot}$) contributions (Fig. 5b). Here, the entropies are normalized with respect to their bulk values (Table 1).

Figure 5b shows that the crossover between the two linear regimes has its roots in the differential changes of the translational and rotational entropy components with $\rho$. The translational entropy $S_{tr}$ remains almost unchanged up to $\rho \approx 150$ mg mL⁻¹, after which it starts to drop linearly. In contrast, the rotational entropy $S_{rot}$ starts to decrease already at low protein concentrations and undergoes a crossover at $\rho \approx 150$ mg mL⁻¹, thereafter continuing to decrease with a slope that is ~2.4 times smaller than at low $\rho$. Hence, it seems that the entropy linked to the rotational motions of water is much more susceptible to the presence of proteins. Moreover, the crossover concentrations of $S_{tr}$ and $S_{rot}$ appear to coincide. A deeper investigation of the underlying effects is intriguing, but is out of the scope of the present work.

To scrutinize the accuracy of the 2PT method, the free energy perturbation (FEP) method[78,79] was used to compute the entropy of bulk water from the free energy of solvation and enthalpy of vaporization.

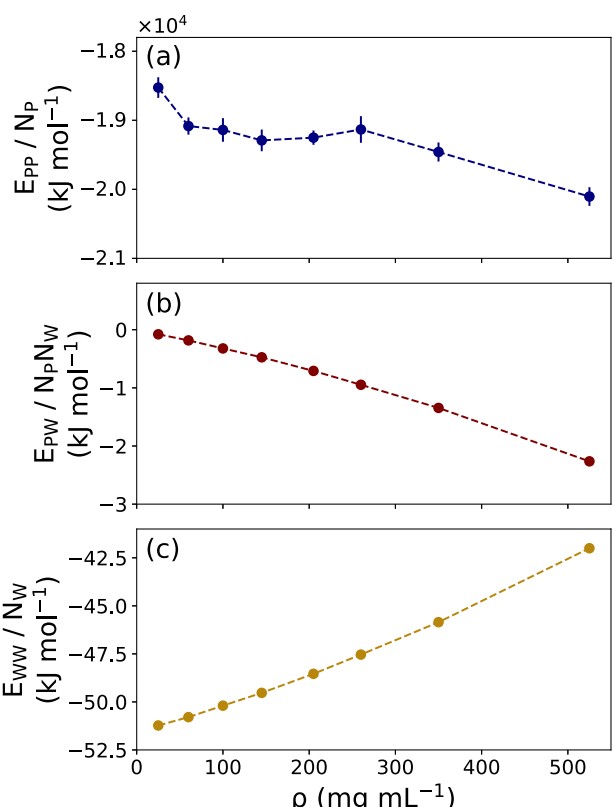

**Fig. 6 | Interaction energies between the different components as a function of protein concentration. a** Protein–protein interaction energy, including intra- and inter-protein contributions. **b** Protein-water interaction energy. **c** Water-water interaction energy (per water molecule). Data in (**a**–**c**) are presented as mean values ± standard deviation (SD) over three repeat simulations (the statistical errors in (**b**, **c**) are smaller than the size of the dots). Source data are provided as a Source Data file.

The total entropy from FEP is 54.4 J mol$^{-1}$ K$^{-1}$, which is very close to the value of 54.7 J mol$^{-1}$ K$^{-1}$ obtained from 2PT (Table 1). A detailed description of the calculations is given in the SI. FEP does not yield the translational and rotational entropies separately, but it is based on a rigorous statistical mechanics framework (that does not rely on an assumed decomposition of the density of states into a solid-like and a gas-like part), and therefore FEP can provide a meaningful and independent validation of the entropy estimates obtained with 2PT.

The property of interest here is the entropy change ($\Delta S$) associated with the formation of a FUS-LCD condensate. The release of water into the dilute surrounding is associated with an increase in entropy that can be calculated by subtracting the entropy of water in a system with protein concentration $\rho$, $S(\rho)$, from that of bulk water, $S(\rho_{dil})$, which is taken as a reference for the dilute phase because the concentration of FUS-LCD in the dilute phase is very low, about 1 mg mL$^{-1}$ (see ref. 48). The total entropy change is then obtained by multiplying with the number of released waters (Eq. (2)). Similarly, the total entropy penalty associated to water retention can be computed by subtracting the entropy of water in a system from that in the protein condensate phase ($\rho_{cond}$ = 350 mg mL$^{-1}$), multiplied by the number of retained water molecules. This is shown in Eq. (12).

$$\Delta S^{rele}(\rho) = \left(S(\rho_{dil}) - S(\rho)\right) \times \Delta N_W^{rele}(\rho)$$
$$\Delta S^{reta}(\rho) = \left(S(\rho_{cond}) - S(\rho)\right) \times \Delta N_W^{reta}(\rho) \tag{12}$$

Note that the numbers of released and retained waters in Eq. (12) are normalized with respect to the number of protein molecules (see Fig. 4). To obtain the total solvent entropy change, the two

contributions are added (Eq. (13))

$$\Delta S^{tot}(\rho) = \Delta S^{rele}(\rho) + \Delta S^{reta}(\rho) \tag{13}$$

However, $\Delta S^{tot}(\rho)$ is not effective in the overall thermodynamics of the process because the part of it originating from water–water interactions is canceled out by a compensating enthalpy term (Eq. (10)). Hence, the entropy bill obtained from Eq. (12) only comprises of the noncanceling term, $T\Delta S_{PW}$, which is obtained by subtracting the calculated water–water interaction energy change from the total solvation entropy term (Eq. (14)).

$$T\Delta S_{PW} = T\Delta S_{solv} - \Delta E_{WW} \tag{14}$$

The other part of free energy originates from the interaction energies. Figure 6 plots the $\rho$-dependence of protein–protein ($E_{PP}$), protein–water ($E_{PW}$), and water–water ($E_{WW}$) interaction energies, normalized by the corresponding numbers of molecules.

Figure 6a shows that $E_{PP}$ decreases with increasing $\rho$. This is a result of favorable protein–protein interactions in the condensate, as have been reported before in experimental and theoretical studies[17,27,80]. The favorable change in $E_{PW}$ (Fig. 6b) arises from the increased fraction of protein-water interactions, mostly through hydrogen bonds, with increasing protein concentration. The simultaneous decrease of water–water HBs (Fig. 3b) accounts for the destabilization of water–water interactions, which results in the increase of $E_{WW}$ with $\rho$ (Fig. 6c).

In addition to entropy, another major thermodynamic contribution is associated to the changes in the interaction energies upon FUS-LCD condensate formation. The above considerations for calculating entropy changes of released and retained water molecules (Eq. (12)) also apply to the changes in energies (Eq. (15))

$$\Delta E^{rele}(\rho) = \left(E(\rho_{dil}) - E(\rho)\right) \times \Delta N_W^{rele}(\rho)$$
$$\Delta E^{reta}(\rho) = \left(E(\rho_{cond}) - E(\rho)\right) \times \Delta N_W^{reta}(\rho) \tag{15}$$

Each of the terms in Eq. (15), $\Delta E^{rele}(\rho)$ and $\Delta E^{reta}(\rho)$, has two contributions, which are the canceling water–water ($\Delta E_{WW}$) and the noncanceling protein–water ($\Delta E_{PW}$) interaction energies. The total change in solvation enthalpy ($\Delta H_{solv}$) is the sum of these two terms (Eq. (8)), although the former term does not influence the change in free energy due to cancellation (Eq. (10)).

The scenario is somewhat different for the changes in protein–protein interaction energies, $\Delta E_{PP}$. Since the reference dilute phase is bulk water (see above), in our analyses we assume that proteins are present only in the condensate phase. Consequently, there are no "released proteins" upon condensate formation and therefore, the total change in protein–protein interaction energy is entirely determined by the protein interactions in the condensate (Eq. (16))

$$\Delta E_{PP} = E_{PP}(\rho_{cond}) - E_{PP}(\rho) \tag{16}$$

The changes in the thermodynamic quantities discussed thus far are plotted in Fig. 7, with separate contributions from retained water, released water, and the total change resulting from the sum of these two individual components (the tug-of-war). Figure 7a shows the total solvation entropy change (Eq. (9), see also Supplementary Table 2). The entropy change for released water favors condensate formation because water molecules explore a larger configuration space in the bulk-like dilute phase. At the same time, the retained water molecules are in a more crowded environment, and thus have a decreased entropy that disfavors phase separation. In this entropy tug-of-war, the contribution of the released water molecules slightly outweighs that of the retained waters, resulting in a slightly negative $-T\Delta S_{solv}$ (Fig. 7a,

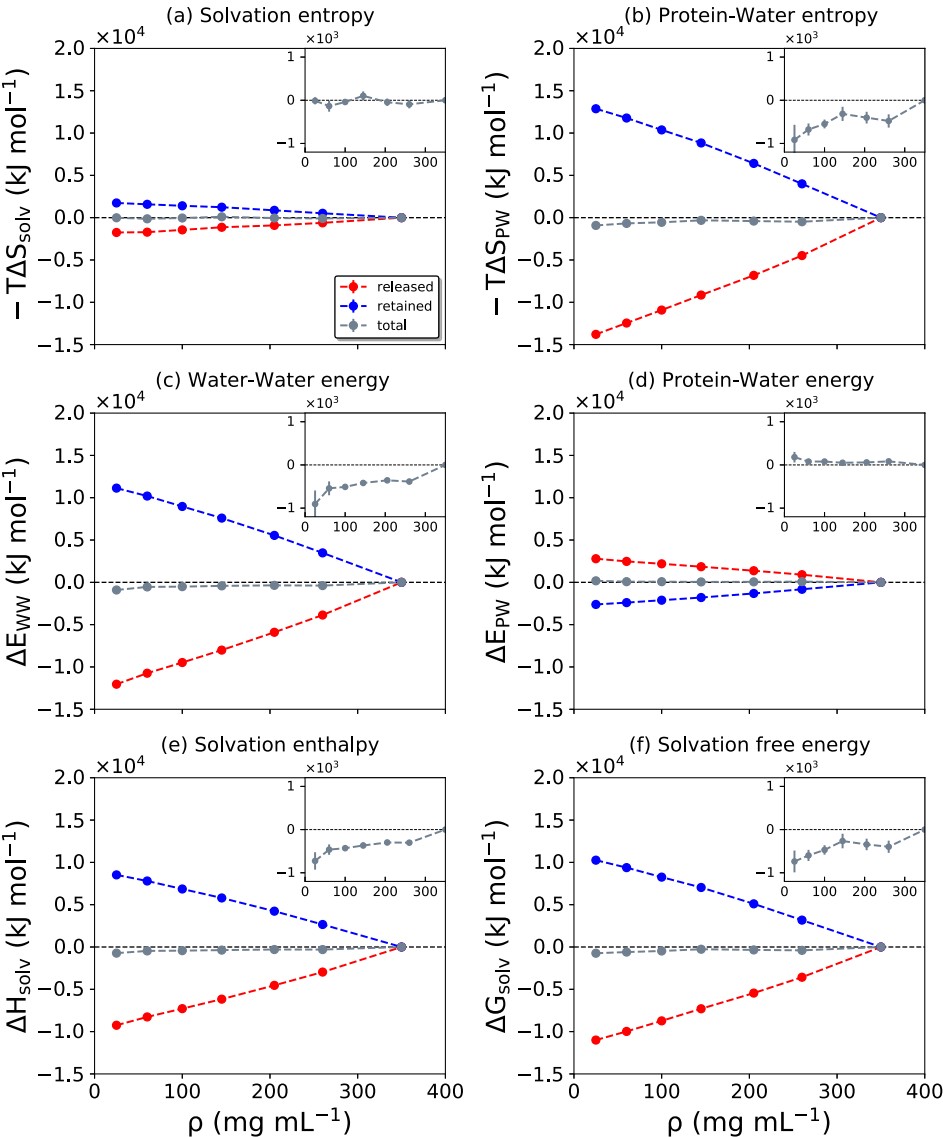

**Fig. 7 | Changes of the solvation-related thermodynamic quantities.** The quantities plotted in (**a**–**f**) are indicated at the top of each panel. The dashed red, blue, gray lines denote the released, retained, and total water contributions, respectively. Data are presented as mean values ± standard deviation (SD) over three repeat simulations. Source data are provided as a Source Data file.

inset), and hence the solvation entropy change favors condensate formation.

The entropic driving forces become clearer from the noncanceling entropy changes originating from the protein–water interactions, Eq. (14) (Fig. 7b and Supplementary Table 3). Although the thermodynamic contributions from released and retained water changes compensate each other to a large extent, the total change in $-T\Delta S_{PW}$ remains favorable (Fig. 7b, inset) and substantially greater than $-T\Delta S_{solv}$. The difference in the magnitudes of $-T\Delta S_{solv}$ and $-T\Delta S_{PW}$ originates from the canceling water–water term, $\Delta E_{WW} = T\Delta S_{WW}$, shown in Fig. 7c and Supplementary Table 4. This $\Delta E_{WW}$ contribution is larger than the noncanceling $\Delta E_{PW}$ energy term (Fig. 7d and Supplementary Table 5).

Interestingly, the released and retained waters behave oppositely in the water–water and protein–water interaction energy contributions (Fig. 7c, d), with the former (water–water) term dominating the total solvation enthalpy change (Fig. 7e and Supplementary Table 6). On approaching the condensate concentration, an increasing fraction of the retained water molecules form protein–water HBs at the cost of water–water HBs (Fig. 3b). Hence,

the interaction energy of the retained waters with the proteins becomes more favorable. Simultaneously, the decrease in the water–water HBs leads to the increase of the interaction energy, that is, unfavorable $\Delta E_{WW}$. At the same time, the released waters gain more water–water contacts at the expense of losing contacts with proteins.

Figure 7f shows that the change in total solvation-free energy is favorable for the formation of FUS-LCD condensates (see also Supplementary Table 7). Hence, in the described tug-of-war between retained and released water molecules, the latter win against the former, resulting in a significant net thermodynamic driving force for the formation of FUS-LCD condensates.

The other part of the total free energy change is not linked to solvation but originates from the proteins. As discussed above in the context of in Eq. (16), this energy contribution is entirely governed by the protein interactions in the condensate (Fig. 8 and Supplementary Table 8). The protein–protein interaction energy, $\Delta E_{PP}$ is attractive at higher concentrations due to favorable contacts between the FUS-LCD molecules. This finding is in line with the fact that FUS-LCD has a large fraction of polar residues, but does not carry a large net charge (−2 in

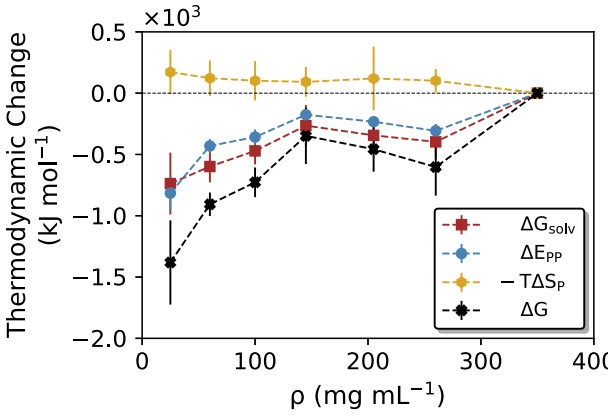

**Fig. 8 | Thermodynamic driving forces of FUS-LCD condensate formation.** The contributions from total solvation-free energy ($\Delta G_{solv}$, brown dashed line), protein–protein interaction energy ($\Delta E_{PP}$, blue dashed line), and protein conformational entropy ($\Delta S_P$, orange dashed line) are plotted together with the resulting total free energy change upon formation of a FUS-LCD condensate ($\Delta G = \Delta G_{solv} + \Delta E_{PP} - T\Delta S_P$, black dashed line), starting from a (hypothetical) homogeneous solution with protein concentration $\rho$. Data are presented as mean values ± standard deviation (SD) over three repeat simulations. Source data are provided as a Source Data file.

our simulations) that would result in repulsive Coulomb interactions at high concentrations.

Finally, the missing thermodynamic contribution is the change in protein conformational entropy upon condensate formation (Eq. (17))

$$\Delta S_P = S_P(\rho_{cond}) - S_P(\rho) \qquad (17)$$

which is expected to be unfavorable due to the increased confinement of the proteins. As explained in "Methods", the protein conformational entropy was estimated using a nearest-neighbor method[81]. Figure 8 (orange curve, see also Supplementary Table 9) shows that indeed, $-T\Delta S_P$ is unfavorable, but it is much smaller than the solvation-free energy and protein–protein interaction energy. As a check, additional control simulations were performed in which two trajectories, one for the 25 mg mL⁻¹ system and one for the 350 mg mL⁻¹ system, were extended to 400 ns and the protein conformational entropies were recalculated based on the final 300 ns of these trajectories. The resulting $-T\Delta S_P$ value of 87.3 kJ mol⁻¹ lies within the error bounds plotted in Fig. 8. The relatively small difference in protein conformational entropy between the dilute and dense phases is in line with the distributions of the radii of gyration of the individual FUS-LCD polypeptide chains in the simulation systems at 25 and 350 mg mL⁻¹ (Supplementary Fig. 2), which suggest that a pronounced degree of protein flexibility is retained also within the condensate.

The accurate estimation of (changes in) protein conformational entropy is notoriously hard because it requires exhaustive conformational sampling[82,83], which is impossible with all-atom MD for an IDP of this size. In light of these limitations, we consider the reported $\Delta S_P$ to be only a rough estimate. However, the main finding that $\Delta G$ is favorable is robust with respect to uncertainties in the protein conformational entropy estimate, as changing $-T\Delta S_P$ even by one order of magnitude does not change the sign of the total free energy change.

Combining all these contributions, the total free energy change is negative (black curve in Fig. 8, see also Supplementary Table 10), in agreement with the experimentally observed spontaneous LLPS of FUS-LCD. The two largest thermodynamic driving forces that favor condensate formation are the noncanceling (protein–water) solvation entropy and the protein–protein interaction energy. These two contributions are comparable in magnitude, showing that solvation effects

are as important as protein interactions for the formation of FUS-LCD condensates.

Finally, since there is no general consensus on the exact FUS-LCD concentration in the condensate, we repeated the analyses using a condensate concentration of 525 mg mL⁻¹ instead of 350 mg mL⁻¹ as a reference. The results are shown in Supplementary Figs. 3–5 and confirm the above conclusions, in that the thermodynamic profiles look very similar. The magnitudes of the individual contributions are larger, as expected due to the more pronounced concentration difference between the condensate and the dilute phase.

To summarize this work, we used large-scale atomistic MD simulations to analyze in molecular detail the thermodynamic driving forces underlying the condensate formation of the intrinsically disordered low complexity domain (LCD) of the human fused in sarcoma (FUS) RNA-binding protein. An understanding of the individual contributions of proteins and water is provided, ultimately yielding a complete thermodynamic picture.

The formation of protein condensates leads to a fraction of the hydration water molecules being released into the dilute phase, which in the case of FUS-LCD has a very low protein concentration and is bulk water-like. The remaining solvent fraction is retained inside the protein condensate, where it is strongly confined by the dense protein environment. The tug-of-war between these two categories of water molecules determines the solvent thermodynamics of the overall process. The retained waters have substantially lower entropy. Furthermore, the hydrogen bonds formed between the retained water molecules and the proteins perturb the extended hydrogen bond network among the water molecules themselves and weaken the water–water interactions. In contrast, the released waters in the dilute phase loose interactions with the proteins and form more water–water H-bonds, and they gain entropy. This latter contribution dominates, and thus the released waters win the tug-of-war, resulting in an entropy-dominated favorable solvation force for FUS-LCD condensate formation.

Favorable protein–protein interaction energy also significantly contributes to the FUS-LCD condensate formation, comparable in magnitude to the total solvation free energy change. This shows that for FUS-LCD, both water and protein are equally important for the thermodynamics of protein condensate formation via LLPS.

By construction, the present study focuses on the formation of FUS-LCD condensates in vitro, that is, aqueous two-phase systems in which the dense condensate is surrounded by a dilute phase. Of course, the actual scenario inside a biological cell is much more complicated. The cell cytoplasm is crowded, and the surrounding of a condensate in a cell, and hence the environment that the released water molecules enter, is far from dilute. Furthermore, the condensate will typically not be comprised only of a single (protein) species, but of multiple different components, including RNA, and it can have internal structural organization[84,85]. Therefore, the magnitudes of the thermodynamic contributions identified in this work should be interpreted rather as upper limits, especially the changes associated with the released water. Nevertheless, we expect that the insights and overall conclusions of this work are transferable to biological cells, at least at a qualitative level, as long as there are distinct regions with substantially different concentrations.

## Methods
### Setup of simulation systems
All simulations were performed using the GROMACS (version 2020.1) molecular dynamics simulation package[86,87]. The Low Complexity Domain (LCD) of the human Fused in Sarcoma (FUS) RNA-binding protein was simulated, which encompasses residues 1 to 163 (out of a total of 526 residues) and has a molar mass of 17.2 kDa. FUS-LCD is an intrinsically disordered protein (IDP) domain for which no experimental high-resolution structure is available.

The initial atomic coordinates of unfolded FUS-LCD were generated with AlphaFold[88,89], which predicts an extended conformation without any secondary structure elements. The first aim was to obtain pre-equilibrated systems at the desired protein concentrations. To accelerate this step, coarse-grained (CG) MD simulations were carried out using the Martini 2.2[90,91] force field. The CG-Martini topology, as generated with the *martinize.py* script available on the Martini website (www.cgmartini.nl), did not contain any elastic network bonds between CG beads. We inserted eight copies of the coarse-grained FUS-LCD in a cubic periodic simulation box with ~27 nm edge length, corresponding to a protein concentration of ~10 mg mL$^{-1}$. In total, 176,366 CG water beads and 16 sodium ions (for neutralization of the charge of the simulation box) were added to the system (1 CG water bead effectively represents 4 water molecules in the Martini force field). Protein–protein Lennard-Jones (LJ) interactions were scaled according to the protocol of ref. 92, which was recently used by Benayad et al. for studying FUS-LCD condensate formation[49]. The well depths of the LJ 6,12 potential were modified with a scaling parameter $\alpha$ according to $\epsilon_\alpha = \epsilon_0 + \alpha(\epsilon_{orig} - \epsilon_0)$, where $\alpha = 0.7$ was found by Benayad et al. to capture the phase separation behavior of FUS-LCD in the CG-Martini simulations. In the above formula, $\epsilon_\alpha$ and $\epsilon_{orig}$ represent the scaled and the original LJ well depths, respectively, and $\epsilon_0 = 2.0$ kJ mol$^{-1}$, which corresponds to the least attractive interaction in the Martini force field[91].

The system was energy minimized and simulated for 1 μs under *NpT* conditions at $T = 300$ K and $p = 1$ bar. Temperature and pressure were maintained using a weak coupling thermostat and barostat[93], respectively. All coarse-grained simulations were performed using the recommended settings[94], which also included the use of a 20 fs time step to integrate the equations of motion.

From the obtained initial equilibrated system, batches of water molecules were removed in a step-wise manner to eventually obtain the desired protein concentrations (25, 60, 100, 145, 205, 260, 350, and 525 mg mL$^{-1}$), see Fig. 2 and Table 2. Before each next water removal step, the systems were further equilibrated under *NpT* conditions for 100 ns. The CG-Martini simulations enable fast equilibration of the protein configurations. However, the approximate and effective nature of the potential energy function and the reduced number of degrees of freedom (e.g., the Martini water model does not have any rotational degrees of freedom) limits the obtainable insights into the protein and solvent contributions to the thermodynamic driving forces. Therefore, the CG simulations were not used for any of the analyses presented in this work, but only to speed up the initial equilibration of the systems. To generate atomistic configurations for the subsequent all-atom MD simulations, the systems were back-mapped to the all-atom level using the *initram.sh* and *backward.py* scripts[95]. The resulting atomistic proteins were solvated using the a99SB-disp water model, which was derived from the TIP4P-D model for usage together with the corresponding a99SB-disp protein force

field[96], which was shown to be a good choice for FUS in a recent systematic comparison of nine force fields[97]. The systems were neutralized by adding sodium ions, and additional 150 mM NaCl was added to mimic physiological ion concentration. The system details are summarized in Table 2. A separate cubic box of neat water, containing 177,382 a99SB-disp water molecules and 150 mM NaCl, with box dimensions of ~17.5 nm, was simulated at 300 K to obtain the reference values for bulk water.

## MD simulation details

All systems were first energy minimized using steepest descent and then equilibrated with harmonic position restraints on the protein backbone for 10 ns (restraining force constants of 1000 kJ mol$^{-1}$ nm$^{-2}$) under *NpT* conditions at $T = 300$ K and $p = 1$ bar. This was followed by further *NpT* equilibration under the same conditions for 100 ns without any position restraints. Thereafter, the systems were equilibrated for 100 ns in the *NVT* ensemble, followed by the final production runs (see below). In these simulations, the velocity rescaling thermostat with a stochastic term[98] and the Berendsen barostat[93] were used to control temperature and pressure, respectively.

Two sets of production runs were carried out. For the vibrational density of states, 20 ps simulations with an output frequency of 4 fs were carried out. For the other calculations, 100 ns simulations with an output frequency of 1 ps were done. Every simulation was repeated three times, starting from different initial configurations. Statistical errors were calculated as standard deviations over these three trajectories. All atomistic MD simulations were done using the leap-frog integrator with a time step of 2 fs. Protein bonds and internal degrees of water molecules were constrained using the LINCS and SETTLE algorithms, respectively. Short-range electrostatic and Lennard-Jones interactions were calculated up to an interparticle distance cutoff of 1.0 nm. Long-range electrostatic interactions were treated with the particle-mesh Ewald technique[99] with a grid spacing of 0.12 nm.

## Analyses

**Interaction energy.** The interaction energies were calculated by post-processing the simulation trajectories using the *mdrun -rerun* routine implemented in GROMACS, with Coulomb and van der Waals distance cut-offs increased to 3.5 nm. Such a large cutoff distance of the pairwise nonbonded interactions was used to minimize the neglect of the lattice part (long-range interactions) in the reruns.

**Entropy of water.** The entropy of water was computed using the the DoSPT implementation[100,101] of the 2-phase-thermodynamics (2PT) method[77,102]. This method was specifically developed to calculate the molar entropies of liquids, and it was shown to yield accurate water entropies in different systems and under a variety of conditions[21,24,103–107]. In 2PT, the spectral density $I(\nu)$ or density of states (DoS) of a liquid is the Fourier transform of the velocity autocorrelation function (VACF), Eq. (18).

$$I(\nu) = \frac{2}{k_B T} \int_{-\infty}^{\infty} e^{i2\pi\nu t} C(t)\, dt \qquad (18)$$

For rigid water molecules, translational ($C_{tr}(t)$) and rotational ($C_{rot}(t)$) VACFs can be treated separately,

$$C_{tr}(t) = m_W \langle \mathbf{v}(\tau)\mathbf{v}(\tau + t) \rangle_\tau$$
$$C_{rot}(t) = \sum_{k=1}^{3} I_k \langle \omega_k(\tau)\omega_k(\tau + t) \rangle_\tau \qquad (19)$$

where $m_W$ is the mass of a water molecule, **v** its center-of-mass velocity vector, $I_k$ the moment of inertia for rotation around the $k$-th principal axis of the water molecule, $\omega_k$ the corresponding angular velocity, and

## Table 2 | Details of simulation systems studied in this work

| FUS conc. (mg mL$^{-1}$) | Box size (nm) | Number of water beads/molecules | | Number of Na$^+$ ions | | Number of Cl$^-$ ions | |
|---|---|---|---|---|---|---|---|
| | | CG | AA | CG | AA | CG | AA |
| 25 | 21.06 | 74,366 | 304,821 | 16 | 870 | 0 | 854 |
| 60 | 15.79 | 30,366 | 125,496 | 16 | 379 | 0 | 363 |
| 100 | 13.21 | 18,366 | 71,268 | 16 | 229 | 0 | 213 |
| 145 | 11.68 | 12,366 | 47,752 | 16 | 165 | 0 | 149 |
| 205 | 10.38 | 8366 | 32,038 | 16 | 122 | 0 | 106 |
| 260 | 9.58 | 6366 | 24,158 | 16 | 101 | 0 | 85 |
| 350 | 8.67 | 4366 | 16,509 | 16 | 79 | 0 | 63 |
| 525 | 7.57 | 2366 | 9273 | 16 | 58 | 0 | 42 |

*CG coarse-grained, AA all-atom.*

$\langle...\rangle_\tau$ is an ensemble average over initial times $\tau$. The 2PT method partitions the total DoS of the liquid into a solid-like ($I^s(\nu)$) and a gas-like ($I^g(\nu)$) contribution,

$$I(\nu) = I^s(\nu) + I^g(\nu) \qquad (20)$$

The solid-like DoS is treated using the harmonic oscillator (HO) model. The diffusive gas-like contribution is described using the Enskog hard sphere (HS) theory (for translation) and the rigid rotor (RR) model (for rotation). The analytically obtained translational and rotational contributions are added to arrive at the total entropy of water (Eq. (21)),

$$S = S_{tr}^{HO} + S_{tr}^{HS} + S_{rot}^{HO} + S_{rot}^{RR} \qquad (21)$$

An advantage of the 2PT approach is that it yields translational and rotational entropy separately. For more detailed descriptions of the method, see the works of Lin et al.[77,102] and the recent review by Heyden[24].

**Protein conformational entropy.** We estimated the conformational entropy of the proteins using the nearest-neighbor (NN) approach[108,109], as implemented in the *PDB2ENTROPY* method introduced by Fogolari et al. that estimates conformational entropies of proteins from probability distributions of torsion angles relative to uniform distributions[81]. The torsion angles are considered for atoms that are within a given cutoff distance (8 Å) from the central atom. Entropy was calculated based on the maximum information spanning tree (MIST) approach[110]. To calculate the conformational entropy difference, $\Delta S_{conf}$, between the condensed and dilute phases, the simulation systems at 350 mg/mL and 25 mg/mL concentration were used. We averaged over all eight FUS-LCD proteins in the simulation boxes and over the three independent repeats.

All the software and codes used in the work are tabulated in Supplementary Table 11.

**Reporting summary**

Further information on research design is available in the Nature Portfolio Reporting Summary linked to this article.

## Data availability

The initial and final coordinates of the MD simulation runs as well as the MD parameter files and force field topology files to be used with GROMACS v.2020.1 are provided as Supplementary Data 1. Source data are provided with this paper.

## Code availability

The computer codes used to run and analyze the MD simulations are available in the public domain. Their accession links are reported in Supplementary Table 11.

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

## Acknowledgements
We thank Matthias Heyden (Arizona State University) for useful discussions. This project received funding from the European Union's Horizon 2020 research and innovation program under the Marie Skłodowska-Curie grant agreement No. 801459-FP-RESOMUS and from the Deutsche Forschungsgemeinschaft (DFG) under Germany's Excellence Strategy-EXC 2033-390677874-RESOLV.

## Author contributions
L.V.S. conceived the study and designed and supervised the project. S.M. carried out all MD simulations, analyzed and interpreted the data, and created the figures, with the help of L.V.S. Both authors wrote the manuscript.

## Funding

## Competing interests
The authors declare no competing interests.
