## [Peer Review File · Nature Communications]

REVIEWER COMMENTS

Reviewer #1 (Remarks to the Author):

Mukherjee and Schafer conduct a careful analysis of model protein condensates to gain insights into the driving forces underlying phase separation of proteins. The paper is a delight to read and is written very clearly despite the technical nature of many topics discussed. This work should motivate others to analyze their data in light of the observations reported here and help figure out how general the findings of this work are.

I have few comments for the authors to consider.

(a) I was a bit surprised by the statement "While the CG-Martini simulations enable fast equilibration, they cannot provide insights into the protein and solvent contributions to the thermodynamic driving forces, which requires all-atom simulations with explicit solvent." I cannot understand how the authors arrived at this conclusion and what the point of CG-Martini simulation with explicit water is. Can the authors conduct a similar analysis of their Martini simulations to demonstrate this? I understand this may require a lot of work that may be outside the scope. In that case, the authors should modify this statement appropriately.

(b) Even though the results presented in the manuscript make sense, I would have liked to see a control to better appreciate the findings. For example, the authors could simulate FUS LC without Tyr residues (mutate it to a polar residue such as Ser) to show that the thermodynamic analysis shows lack of driving force for condensate formation. Again, this may involve a lot of extra work, so I leave to the author's discretion how to address it. I believe it will make the paper much stronger.

(c) The authors should expand on the statement "...the best choice for this protein in a recent systematic comparison of nine force fields". In what sense was this force field deemed best?

(d) The authors may want to think carefully about Fig. 3a data. I see no change happening within a significant concentration range as opposed to a perceived linear decrease.

(e) I didn't fully understand eq. (14). May be authors can add some explanation for non-experts.

Reviewer #2 (Remarks to the Author):

Mukherjee and Schaefer present a very timely and important study of the molecular driving forces of liquid-liquid phase separation. This is - to my knowledge - the first atomistic molecular dynamics investigation of the relative importance of protein-protein interactions and protein solvation for liquid-liquid phase separation. The study shows that solvent entropy changes and increased protein-protein interaction are the main drivers of the condensation of FUS LCD. As condensates formed some water molecules are released from the protein condensate. These released water molecules gain entropy. Other water molecules are retained close to the protein and are highly confined due to condensation and have substantially lower entropy. The overall contribution of solvent entropy is on the order of the favourable protein-protein interactions.

Establishing an atomistic perspective on protein phase separation and its drivers is a significant advance. As such this a really important study that will complement polymer theory approaches and experimental investigations. As such the manuscript warrants publication in Nature Communications and I strongly support its acceptance after considering a few relatively minor issues discussed below, which can further enhance the impact of this work.

Detailed Comments

1) I really appreciate that the authors validated their solvent entropy calculation by independent Free energy perturbations calculations, which is very impressive and illustrates the high level of this atomistic molecular dynamics study.

2) The authors may consider running one additional simulation of a dense system and a dilute system to check their estimate of the protein conformational entropy. Of course there is a limit of what is possible, so I leave it to the authors to decide on whether they follow this suggestion.

3) Is there a tendency for the proteins to collapse (is R_g stable overtime)? For this, it may make sense to extend one simulation of a dilute and dense FUS solution to ca. 1 μ s. Likely, a possible tendency to over-estimate protein-protein in atomistic force fields would not really change the conclusions. Presumably, insufficient equilibration is a larger issue here than any slight mis-balance in the atomistic force field.

4) Again considering R_g values, how flexible are the proteins in the condensate?

5) Recent work from Heinz Grubmüller expands on Ben Naim's theorem <https://arxiv.org/abs/2306.09392>. Does this apply here? Maybe the authors could briefly comment on these new results and whether these apply to the present study or not.

6) While all-atom simulations of condensates of disordered proteins are still a relatively new field, any list of simulation studies is by necessity incomplete. What about referring interested readers to a recent perspective by e.g. J. Mittal? Again, this is a very minor issue that I only bring up since the quality of the manuscript is very high overall.

7) The long-term impact of this study could be enhanced further if the authors make some of their scripts or tools available via a public repository. I understand if that's not possible may be the tools are not so easy to use at this current state and would be used incorrectly by non-experts.

Reviewer 1

Comment 1. Mukherjee and Schafer conduct a careful analysis of model protein condensates to gain insights into the driving forces underlying phase separation of proteins. The paper is a delight to read and is written very clearly despite the technical nature of many topics discussed. This work should motivate others to analyze their data in light of the observations reported here and help figure out how general the findings of this work are. I have few comments for the authors to consider.

Reply. We thank the reviewer for these positive and encouraging comments.

Comment 2. I was a bit surprised by the statement "While the CG-Martini simulations enable fast equilibration, they cannot provide insights into the protein and solvent contributions to the thermodynamic driving forces, which requires all-atom simulations with explicit solvent." I cannot understand how the authors arrived at this conclusion and what the point of CG-Martini simulation with explicit water is. Can the authors conduct a similar analysis of their Martini simulations to demonstrate this? I understand this may require a lot of work that may be outside the scope. In that case, the authors should modify this statement appropriately.

Reply. MARTINI water is a Lennard-Jones fluid (no electrostatic interactions), in which 4 atomistic water molecules are mapped onto a single CG bead. Our present study reveals in detail how the solvent entropy in the condensate is modulated as a result of intricate changes of both the translational and the rotational (orientational) motions of the water molecules, which are part of an extended H-bonded network and thus participate in collective dynamic modes involving many waters. Such dynamics cannot be present in MARTINI water, which does not have any H-bonds nor any rotational degrees of freedom. Furthermore, as a result of the reduced number of degrees of freedom, both the CG water and the CG protein have a much lower entropy than their atomistic counterparts (and, correspondingly, also a different energy in order to – hopefully – arrive at a similar free energy landscape). Taken together, the MARTINI model cannot be expected to yield a realistic picture of the entropy and how it changes upon condensate formation, and we would therefore prefer to refrain from discussing these aspects in our manuscript.

Concerning the question "what the point of CG-MARTINI simulations is", we would like to stress that these were only used in the present work to speed up the initial equilibration ("mixing") of the proteins in our simulation systems, so that the all-atom simulations could be started from properly pre-equilibrated configurations.

To explain these aspects more clearly, we have modified the text on page 7 in the Methods section, which now reads:

"The CG-Martini simulations enable fast equilibration of the protein configurations. However, the approximate and effective nature of the potential energy function and the reduced number of degrees of freedom (for example, the Martini water model does not have any rotational

degrees of freedom) renders it difficult to obtain insights into the protein and solvent contributions to the thermodynamic driving forces. Therefore, the CG simulations were not used for any of the analyses presented in this work, but only to speed up the initial equilibration of the systems.”

Comment 3. Even though the results presented in the manuscript make sense, I would have liked to see a control to better appreciate the findings. For example, the authors could simulate FUS LC without Tyr residues (mutate it to a polar residue such as Ser) to show that the thermodynamic analysis shows lack of driving force for condensate formation. Again, this may involve a lot of extra work, so I leave to the author's discretion how to address it. I believe it will make the paper much stronger.

Reply. Thank you for the suggestion. We agree that such control would indeed be valuable. However, as the reviewer acknowledges, repeating all simulations, for example with an (artificial) FUS-LCD variant in which all tyrosine residues are changed to serines, would be very time-consuming given the very large system sizes and long simulation times needed for protein condensates (the results included in the present manuscript required more than 1.5 years (real time) of parallel MD simulations on a supercomputer).

[Redacted]

[Redacted]

Comment 4. The authors should expand on the statement "...the best choice for this protein in a recent systematic comparison of nine force fields". In what sense was this force field deemed best?

Reply. In the paper *J. Chem. Theory Comput.* **2023**, *19*, 3721-3740., Sarthak *et al.* have compared 5 all atom force fields to study the radius of gyration (Rg) of FUS (single protein in the simulation box, no condensate) and have compared their results with experimental observations. According to their report, the traditional force fields such as AMBER ff14SB and CHARMM36m along with the TIP3P water model result in unnaturally compact FUS configurations, resulting in reduced internal dynamics. Replacing the water force field with other 4-point models (OPC (ff19SB) or TIP4P(ff03ws)) results in less compact protein conformations but still not close to that observed in experiments. However, the AMBER ff99SB-disp force field that we have used in this work yields Rg values in agreement with experiments, and the dynamic nature of the protein chains are retained.

Comment 5. The authors may want to think carefully about Fig. 3a data. I see no change happening within a significant concentration range as opposed to a perceived linear decrease.

Reply. We agree with the reviewer. In Fig. 3a, we had included a linear fit merely as a visual guide to the eye (this was also stated in the figure caption). However, we fully agree with the reviewer in that the actual data do not seem to support the idea of a simple linear relationship

here, and we are also not aware of a theoretical model that would suggest so. Therefore, we have removed the linear fit from the plot and adapted the figure caption text accordingly.

Comment 6. I didn't fully understand eq. (14). May be authors can add some explanation for non-experts.

Reply. Thank you for bringing this up. We have added some text that puts the equation into context on p. 19 (below Eq. 14).

Eq. 14 states the exact entropy-enthalpy compensation for solvent-solvent interactions. When a solute is present in a solution, the total solvation enthalpy arises from two distinct contributions: the solute-solvent interactions and the solvent-solvent interactions. Similarly, the solvation entropy can (formally) also be expressed as a sum of two analogous terms. However, the thermodynamic changes associated with solvent-solvent interactions lead to an exact cancellation of the entropy and enthalpy terms. Consequently, the total solvation free energy is effectively represented solely by the non-canceling solute-solvent interaction enthalpy and entropy terms.

The idea of entropy-enthalpy compensation was initially shown by Ben Naim in his work on hydrophobic interactions involving solvents. Subsequently, Ben-Amotz et al. and Karplus et al. have substantiated the validity of this concept. In the context of our study, water serves as the solvent, and the protein is the solute. In Eq. 14, ΔE_{WW} and $T\Delta S_{WW}$ represent the enthalpy and entropy changes associated with water-water interactions in the system, and they cancel each other (as originally proven by Ben-Naim).

Reviewer 2

Comment 1. Mukherjee and Schaefer present a very timely and important study of the molecular driving forces of liquid-liquid phase separation. This is - to my knowledge - the first atomistic molecular dynamics investigation of the relative importance of protein-protein interactions and protein solvation for liquid-liquid phase separation. The study shows that solvent entropy changes and increased protein-protein interaction are the main drivers of the condensation of FUS LCD. As condensates formed some water molecules are released from the protein condensate. These released water molecules gain entropy. Other water molecules are retained close to the protein and are highly confined due to condensation and have substantially lower entropy. The overall contribution of solvent entropy is on the order of the favourable protein-protein interactions.

Establishing an atomistic perspective on protein phase separation and its drivers is a significant advance. As such this a really important study that will complement polymer theory approaches and experimental investigations. As such the manuscript warrants publication in Nature Communications and I strongly support its acceptance after considering a few relatively minor issues discussed below, which can further enhance the impact of this work.

Reply. We thank the reviewer for the positive comments.

Comment 2. I really appreciate that the authors validated their solvent entropy calculation by independent Free energy perturbations calculations, which is very impressive and illustrates the high level of this atomistic molecular dynamics study.

Reply. Thank you for the encouraging comment.

Comment 3. The authors may consider running one additional simulation of a dense system and a dilute system to check their estimate of the protein conformational entropy. Of course there is a limit of what is possible, so I leave it to the authors to decide on whether they follow this suggestion.

Reply. We understand that the reviewer suggests additional controls/checks of the statistical convergence of the protein conformational entropy estimates. Indeed, as is discussed in the text, these are subject to large uncertainties due to the (well known) sampling challenges. The values given in the original manuscript result from three independent simulation runs, with different initial atomic configurations. The statistical errors resulting from these 3 repeats were given as error bars in Fig. 8 in the original manuscript.

As a check, we had run additional analyses of our simulations of the higher protein concentration (525 mg/mL instead of 350 mg/mL). These results are shown in the SI (Fig. S4) and are in line with the ones shown in Fig. 8, confirming that even at this higher protein concentration, the contribution to the free energy change due to changes of the protein conformational entropy is relatively small compared to the other (major) thermodynamic contributions, which are water entropy and protein-protein interaction energy.

Furthermore, we followed the suggestion by the reviewer (see comment 4 below) and extended the simulations of the dilute (25 mg/mL) and dense (350 mg/mL) systems to 400 ns (reaching 1 microsecond was not possible within the given timeframe, especially not for the 25 mg/mL system with 1.2 M atoms). The differences between the original protein conformational entropy values shown in Fig. 8 of the manuscript (which were based on the three 100 ns repeats) and the new $-T\Delta S$ values (based on the last 300 ns of the extended 400 ns simulation, just 1 repeat) are 165.3 kJ/mol and 87.3 kJ/mol respectively. That is, the $-T\Delta S$ value from the extended simulation is 1.9 times smaller than the original (previous) value. This result confirms our conclusion that the $-T\Delta S$ protein conformational entropy contribution to the overall free energy change upon LLPS is much smaller (by about 1 order of magnitude) than the other, dominant contributions. In line with this finding, additional analyses of the radii of gyration (see below) confirm that the dynamic nature of the FUS-LDC chains is retained also within the condensate, and hence the conformational freedom of the proteins is not strongly reduced within the dense phase. These additional controls are mentioned on p. 27 of the revised manuscript.

Comment 4. Is there a tendency for the proteins to collapse (is R_g stable over time)? For this, it may make sense to extend one simulation of a dilute and dense FUS solution to ca. 1 μ s. Likely, a possible tendency to over-estimate protein-protein in atomistic force fields would not really change the conclusions. Presumably, insufficient equilibration is a larger issue here than any slight mis-balance in the atomistic force field.

Reply.

As mentioned in the previous reply, we extended the simulations of the 350 mg/mL and of the 25 mg/mL systems (to 400 ns). The above figure shows the time traces of the radii of gyration (R_g) of each of the eight proteins in both the most dilute solution (25 mg/mL) and the condensate (350 mg/mL). Apparently, there is no slow collapse (drift in R_g) in any of these trajectories over the time period of the simulations (apart from the green curve in the 25 mg/mL system maybe, which shows a pronounced fluctuation (but towards larger R_g values, not smaller ones!) but eventually returns to a value that is close to the one at the beginning of the simulation).

Hence, we conclude that the proteins do not collapse within our simulation time. The Amber99sb-disp force field used in this work was particularly developed to better describe both folded and disordered protein states. The intrinsic disorder in the individual FUS-LCD chains and the dynamic disorder in the condensate phase as a whole keeps the proteins chains flexible, see also answer to the next point below.

Comment 5. Again considering R_g values, how flexible are the proteins in the condensate?

Reply.

The figure above, which we added to the SI (new Fig. S2), shows the R_g distributions for each of the 8 proteins in the dilute and the dense solutions. The proteins have broad distributions of R_g in the 25 mg/mL solution, supporting the notion that they are indeed quite flexible. In the dense solution (350 mg/mL), the protein chains are somewhat more compact (but not that much), but overall the distributions clearly suggest that the protein chains still retain some degree of flexibility.

We have added a short description to the text (on p. 27 of the revised manuscript).

As an additional, complementary analysis, we computed the time-correlation functions of R_g , thus obtaining insights into the differences in the dynamic time scales associated with the global motions of the protein chains (as are reflected in changes of R_g). As shown in the figure below, the TCF decays somewhat slower at 350 mg/mL (as expected), but the time constants do not differ that much between 25 and 350 mg/mL – both systems clearly can be classified as “dynamic” on the nanosecond time scale. As the precise nature of the protein dynamics, and their attenuation in the condensate, are not the main focus of the present manuscript, we would prefer not to include these results here but leave them for a follow-up study.

Comment 6. Recent work from Heinz Grubmüller expands on Ben Naim's theorem <https://arxiv.org/abs/2306.09392>. Does this apply here? Maybe the authors could briefly comment on these new results and whether these apply to the present study or not.

Reply. We thank the reviewer for pointing out this important reference, which we now cite in the revised version of our manuscript (Ref. 105). In their recent work, Heinz and Grubmüller have clarified on entropy-enthalpy compensation. The apparent confusion arises from the definition of solvent-solvent interaction entropy. According to Ben Naim's theorem, which was subsequently proven by Ben-Amotz et al. and Yu and Karplus, the solvent-solvent interaction entropy contribution is exactly cancelled out by an analogous enthalpy term. However, Heinz and Grubmüller showed that multibody interactions between solvent molecules, accessible via a mutual information expansion (MIE), make sizeable contributions. In this context, it is important to understand that the solvent-solvent interaction entropy is different from the multibody solvent interactions as defined by MIE. In the present work, we have not used MIE but the 2PT method, which does not yield the multibody mutual information explicitly. Hence, we do not run into this apparent confusion. However, if analysed in terms of multibody correlations, the conclusions by Heinz and Grubmüller would also apply in our study as well.

However, given that our manuscript is indeed already quite technical (as is also mentioned by Reviewer 1), and – more importantly – that the (indeed very relevant!) conclusions of Heinz and Grubmüller do not directly concern our present study, we would prefer not to include a discussion of their work in the present manuscript. The references are cited in our manuscript (Refs. 102 - 105), so that the interested reader can refer to the literature.

Comment 7. While all-atom simulations of condensates of disordered proteins are still a relatively new field, any list of simulation studies is by necessity incomplete. What about referring interested readers to a recent perspective by e.g. J. Mittal? Again, this is a very minor issue that I only bring up since the quality of the manuscript is very high overall.

Reply. We have added the recent Curr. Opin. article by J.-E. Shea, R. B. Best and J. Mittal (Ref. 56 in the revised manuscript).

Comment 8. The long-term impact of this study could be enhanced further if the authors make some of their scripts or tools available via a public repository. I understand if that's not possible may be the tools are not so easy to use at this current state and would be used incorrectly by non-experts.

Reply. We thank the reviewer for the suggestions. The codes used in this work are available in the public domain (see Table below, which was also added to SI (Table S11)), and they are referred to in the Methods section of the text. However, if a researcher has any confusion with their use, we can clarify that via email.

Name of the code	Function	Source
DoSPT	Calculation of water entropy	www.dospt.org
pdb2entropy	Calculation of protein conformational entropy	https://github.com/federico-fogolari/pdb2entropy
gmx mdrun -rerun gmx energy gmx hbond gmx hydorder gmx gyrate	Calculation of interaction energies, number of hydrogen bonds, tetrahedral order parameter Radius of gyration	https://www.gromacs.org/